# Does Incretin Agonism Have Sustainable Efficacy?

**DOI:** 10.3390/cells13221842

**Published:** 2024-11-07

**Authors:** Sok-Ja Janket, Miyo K. Chatanaka, Dorsa Sohaei, Faleh Tamimi, Jukka H. Meurman, Eleftherios P. Diamandis

**Affiliations:** 1Retired Research Associate Professor, Boston University Goldman School of Dental Medicine, Boston, MA 02118, USA; skjanket@bu.edu; 2Department of Laboratory Medicine and Pathobiology, University of Toronto, Toronto, ON M5S 1A8, Canada; miyo.chatanaka@mail.utoronto.ca; 3M.D., C.M. Candidate 2026, Faculty of Medicine and Health Sciences, McGill University, Montreal, QC H4A QT2, Canada; dorsa.sohaei@mail.mcgill.ca; 4Department of Restorative Dentistry, College of Dental Medicine, Qatar University, Doha P.O. Box 2713, Qatar; fmarino@qu.edu.qa; 5Department of Oral and Maxillofacial Diseases, Helsinki University Hospital and University of Helsinki, FI-00290 Helsinki, Finland; jukka.meurman@helsinki.fi; 6Lunenfeld-Tanenbaum Research Institute, Sinai Health System, Toronto, ON M5T 3L9, Canada

**Keywords:** glucagon-like peptide 1, glucose-dependent insulinotropic polypeptide, satiety, proopiomelanocortin (POMC), neuropeptide Y (NPY), agouti-related peptide (AgRP), GABAergic neurons

## Abstract

Recent clinical trials using synthetic incretin hormones, glucagon-like peptide 1 (GLP-1), and glucose-dependent insulinotropic polypeptide (GIP) receptor agonists have demonstrated that these treatments ameliorated many complications related to obesity, emphasizing the significant impact of body weight on overall health. Incretins are enteroendocrine hormones secreted by gut endothelial cells triggered by nutrient ingestion. The phenomenon that oral ingestion of glucose elicits a much higher insulin secretion than intra-venous injection of equimolar glucose is known as the incretin effect. This also alludes to the thesis that food intake is the root cause of insulin resistance. Synthetic GLP-1 and GIP agonists have demonstrated unprecedented glucoregulation and body weight reduction. Also, randomized trials have shown their ability to prevent complications of obesity, including development of diabetes from prediabetes, reducing cardiovascular disease risks and renal complications in diabetic patients. Moreover, the benefits of these agonists persist among the patients who are already on metformin or insulin. The ultimate question is “Are these benefits of incretin agonism sustainable?” Chronic agonism of pancreatic β-cells may decrease the number of receptors and cause β-cell exhaustion, leading to β-cell failure. Unfortunately, the long-term effects of these drugs are unknown at the present because the longest duration in randomized trials is 3 years. Additionally, manipulation of the neurohormonal axis to control satiety and food intake may hinder the long-term sustainability of these treatments. In this review, we will discuss the incretins’ mechanism of action, challenges, and future directions. We will briefly review other molecules involved in glucose homeostasis such as amylin and glucagon. Amylin is co-expressed with insulin from the pancreas β-cells but does not have insulinotropic function. Amylin suppresses glucagon secretion, slowing gastric emptying and suppressing the reward center in the central nervous system, leading to weight loss. However, amylin can self-aggregate and cause serious cytotoxicity and may cause β-cell apoptosis. Glucagon is secreted by pancreatic α-cells and participates in glucose homeostasis in a glucose-dependent manner. In hypoglycemia, glucagon increases the blood glucose level by glycogenolysis and gluconeogenesis and inhibits glycogenesis in the liver. Several triple agonists, in combination with dual incretins and glucagon, are being developed.

## 1. Introduction

By the definition of La Barre [1], an incretin should satisfy two criteria: (a) it must be released by nutrients, especially carbohydrates and; (b) it must stimulate insulin secretion in the presence of elevated blood glucose levels. Among the potential incretin candidates such as gastrin, secretin, cholecystokinin (CCK), vasoactive intestinal peptide (VIP), peptide histidine isoleucine (PHI), gastrin-releasing peptide (GRP), entero-glucagon, glucagonlike peptides (GLPs), and glucose-dependent insulinotropic polypeptides (GIP), GLP-1 and GIP are the most important and most researched incretins.

GIP is secreted from K cells in the small intestine by proteolysis of pre-pro-GIP [2], while GLP-1 is expressed in the intestinal L cells and derived from pro-glucagon through differential proteolytic cleavage [3]. Both GIP and GLP-1 act on G protein-coupled receptors on islet β cells to stimulate insulin secretion [4]. Although both stimulate insulin secretion, there are subtle differences between these two molecules. GLP-1 suppresses glucagon secretion, but GIP stimulate glucagon secretion. Currently, there is much interest in adding glucagon to incretins. However, glucagon has some untoward side effects and caution must be taken. Please refer to our Section 4. Potential Adverse Effects. Although at a preliminary stage, triple agonists might have more serious adverse events such as heart rate increase and arrythmia [5]. AstraZeneca discontinued further research after a phase III trial with the triple agonist, cotadutide, which consists of glucagon, oxyntomodulin (OXM), and the GLP-1 (liraglutide) agonist. The exact reason for the discontinuation has not been published.

GLP-1 suppresses appetite via neuronal inputs, while GIP’s role in appetite is conflicting. Although neurons expressing the GIP receptors are identified in the brain of both mice and humans, animal studies report GIP’s involvement in appetite suppression [6,7,8] but human studies report almost no effects of GIP on appetite suppression [9,10,11]. We believe that appetite control involves numerous hormones and neurons, so focusing on just one neuron or molecule, as was done in most animal studies, might not provide a complete picture. Moreover, each molecule can be stimulated or inhibited. Thus, the end results may have many possible outcomes.

The clinical successes of the incretin trials utilizing GLP-1 receptor agonists (GLP-1ras) and GIP receptor agonists (GIPras) have been nothing short of “astonishing” in weight reduction [12,13], and in glucoregulation in type 2 diabetes (T2D) [14] and prediabetes [15]. Further, incretin efficacy is reported in the prevention of cardiovascular complications [16], and as delaying the complication of diabetes-related kidney dysfunction [17]. The most recent trials utilizing dual (a combination of GLP-1 and GIP) or triple receptor agonists demonstrated more robust weight reduction and glycemic control than GLP-1 receptor agonists (GLP-ras) alone [18,19,20,21,22].

We have summarized some key randomized trial results showing these benefits in Table 1. These are not a complete collection of references but only essential trials that demonstrated the superiority of incretin agonism compared to traditional diabetes treatments or other GLP-1 ras.

The two incretins have shared functions as well as divergent functions. They share the insulinotropic function but, when combined, they seem to have additive action. GIP has a more powerful insulinotropic action and is responsible for 44% of the total insulin responses, and GLP-1 contributes 22% [33]. We review the role of incretins and other molecules involved in glucose homeostasis and weight management in the next section.

### 1.1. GLP-1 and GIP Mechanism of Action

When GLP-1 and GIP bind to their cognate receptors, they stimulate insulin secretion from the pancreatic β-cell through the incretin effects [34]. Both incretin-related receptors (GIPr) and GLP-1r belong to the class B family of 7-transmembrane G protein-coupled receptors (GPCR). GIP and GLP-1 share insulinotropic actions, but other functions may be divergent. For example, GIP stimulates glucagon secretion from pancreatic α-cells in hypoglycemia in healthy persons but, in T2D, the glucagonotropic function of GIP is dysregulated [35]. While GLP-1 is known to inhibit glucagon secretion possibly via somatostatin [36], delay gastric emptying, and suppress food intake, GIP does not appear to slow gastric emptying [37]. Reports regarding GIP’s role in appetite suppression are also conflicting: animal studies and in vitro studies have reported GIP’s role in appetite suppression [6,7], while human studies have not [9,11]. Another reason for combining GLP-1 and GIP agonists is that GIP has an anti-emetic function and counteracts nausea and vomiting evoked by GLP-1 ra [38].

### 1.2. Amylin Mechanism of Action

Amylin is a 37-amino acid peptide hormone co-expressed with insulin from the pancreas β-cells, but it does not have insulinotropic function. For this reason, amylin is not considered an incretin [39]. Amylin reduces endogenous glucose production by suppressing glucagon secretion, slows gastric emptying, and suppresses CNS reward centers, leading to weight loss. However, amylin, as the name ‘islet amyloid polypeptide’ suggests, can self-aggregate and cause endoplasmic reticulum stress, serious cytotoxicity, and may cause β-cell death [40]. Therefore, the clinical utility of amylin is very limited.

### 1.3. Glucagon Mechanism of Action

Glucagon is a peptide with 29 amino acids and is secreted by α-cells of the pancreas. It is not an incretin, but intimately participates in glucoregulation and body weight management. Glucagon receptors (GCGR) are expressed in many organs, including the liver, kidney, and heart, among other organs. The main biological function of glucagon is counter-balancing insulin in hypoglycemia. It is also involved in hepatic lipid and amino acid metabolism. Additionally, glucagon is known to enhance satiety and suppress food intake, and it has become an attractive molecule for body weight management [41]. Glucagon also promotes lipogenesis and ketone body formation from non-carbohydrate energy. In times of high energy demand, glucagon converts fatty acids to acetyl-coenzyme A via β-oxidation in the liver [42]. Additionally, glucagon activates the signaling pathway to inhibit hepatic *de novo* lipogenesis and prevents the onset of hepatic steatosis [43]. Although the activation of GCGR is involved in the body weight-lowering action of OXM, its involvement in weight loss appears to be redundant with the GLP-1 agonism. In T2D, however, glucagon regulation is abnormal [44]. The newly developed triple agonist, retatrutide, contains the GLP-1, GIP, and GCGR agonist. However, adding three powerful molecules together may beget a hidden danger in long-term usage. Let us recapitulate the benefits of incretins.

(a)Increase insulin secretion and improve glycemic control. Some studies have also reported insulin sensitivity. However, insulin sensitivity was reported to be associated with inflammation arising from obesity [45,46].(b)Suppress food intake by slowing gastric emptying and increasing satiety, which lead to weight loss [11]. Interestingly, semaglutide did not delay gastric emptying (GE) when assessed using paracetamol absorption in a recent trial [47]. The reason for the conflicting results may be that older studies used the dye dilution method to quantify GE [48]. The dye dilution method is not as precise as the validated GE assessment method, scintigraphy, while recent studies used paracetamol which is known to have comparable precision to scintigraphy [49]. Although the recent study reporting the null effects of semaglutide on GE used a more precise method, it should be noted that the authors were employees of Novo Nordisk, the maker of semaglutide. Murine models of GLP-1 did not slow effects on gastric emptying [50]. These reports indirectly suggest that manipulating the neurohormonal axis using incretins may be the cause of weight loss. The same pathway also reduces the craving for alcohol intake. Unfortunately, when incretin’s manipulation of neuronal pathways is terminated, approximately 2/3 of the lost weight was regained [25,51].(c)May potentiate functional β-cell regeneration in animal or in vitro studies [52]. However, clear evidence of functional β-cell regeneration or proliferation in humans is lacking [36].(d)May prevent bone fractures [53]. However, not all studies reported beneficial effects of incretin agonism on bone mineral density (BMD). Two randomized trials reported that exercise might be a better option for BMD than GLP-1 agonism [54,55]. Further well-conducted studies are needed.

Several reports suggest that relative hyperglucagonemia contributes to fasting and postprandial hyperglycemia in T2D, and glycemic control may be achieved by blocking glucagon action [56]. Moreover, numerous reports suggest the deleterious effects of GLP-1 ra as well as glucagon on heart rate and other cardiac functions [57,58,59,60]. Please refer to Section 4 for further reading.

### 1.4. Long-Term Efficacy of Incretin Agonism

As we stated in the abstract, the long-term effects of incretin agonism are unknown at present. This is because there are no long-term clinical trials, and the longest-term randomized trial has been 160 weeks (40 months). As the GLP-1ras have become the leading treatment for obesity and diabetes, their long-term efficacy is a critical issue given the high cost and potential harm. We, as well as other scientists, have raised the same issue, that there is no evidence for the long-term sustainable efficacy of GLP-1ras [61].

Although several extension studies have been published, these are not randomized trials. Rather, they are bias-prone observational studies. Nevertheless, these extension studies gave us a glimpse of the long-term effects of incretin agonism. The longest extension study is DURATION-1, where exenatide was given once weekly (QW) subcutaneously and followed for 7 years [62]. The drop-out rate of this extension study was 59% and this is almost three times higher than the acceptable 20% drop-out rate [63]. The weight loss was attenuated to just 3.9% in the 7th year. This is quite important, because Moll et al. reported that body weight loss of at least 10% is necessary to observe a salient benefit over harm, while a 5% weight loss does not show any net benefit [64]. The harms in these analyses are mostly gastrointestinal (G-I) adverse effects. In a meta-analysis, Zhang et al. observed that the adverse G-I effects were due to neurohormonal suppression of the appetite center in the brain [65].

Additionally, the clear trend of attenuation in efficacy was observed with increasing study duration. Mean HbA1c reduction was 1.7%, 1.6%, and 1.5% at the 2nd, 5th, and 7th year, respectively. In the 3rd year, only 28% and, in the 7th year, 46%, of the study subjects achieved HbA1c 7%. Seemingly improving HbA1c levels in the 7th year were due to the high drop-out rate: in the 3rd year, the cohort had 194 subjects, which decreased to 122 in the 7th year. The DURATION-1 authors [62] stated that some CVD markers improved in DURATION -1. However, this is irrelevant to GLP-1 agonism because the study participants were taking several CVD medications in addition to GLP-1 agonists. Thus, improved CVD markers cannot be attributed to GLP-1 agonism. One positive comment was that the adverse effects were similar to those in the randomized portion of the study [62].

## 2. Role of Incretins in the Neurohormonal Axis of Appetite Control

Homeostatic feeding is a mechanism where energy for basic metabolic processes and survival are obtained, while hedonic feeding is driven by sensory perception or pleasure [66]. Homeostatic feeding is tightly controlled by many molecules, hormones, and neuronal elements. These include sensing nutrients in the central nervous system (CNS), integrating afferent stimuli, reflecting the energy balance, and adjusting subsequent food intake [67]. The incretin effect is largely mediated by neuroendocrine actions and is correlated with the size of the meal [39].

In the neuroendocrine control of food intake, the brainstem and hypothalamus are the core CNS areas because they receive, convey, and integrate peripheral signals. The area postrema (AP) and nucleus tractus solitarius (NTS) in the brainstem convey the peripheral signals, consisting of nutrients, hormones, and vagal afferent inputs, to the arcuate nucleus (ARC) of the hypothalamus. The ARC contains both orexigenic and anorexigenic neurons. The former expresses neuropeptide Y (NPY) and agouti-related peptides (AgRP) and the latter expresses pro-opiomelanocortin (POMC) and the cocaine/amphetamine-related transcript (CART). They collectively process the received information and regulate eating and attain energy homeostasis [68,69].

POMC cells activate the melanocortin 4 receptor (MC4R), expressing neurons in the paraventricular nucleus of the hypothalamus (PVH) and other brain regions, thereby inhibiting food intake and increasing energy expenditure. The MC4R gene is involved in the brain’s regulation of appetite and weight. Conversely, NPY/AgRP neurons antagonize these effects [70].

Also, leptin and serotonin are involved in regulating energy balance, appetite, and bone mass. Although both leptin and serotonin depolarize POMC neurons [71,72], there is a distinct selectivity in the responsive neurons. Namely, serotonin-responsive POMC neurons are not activated by leptin. Also, these two groups of neurons are anatomically segregated: leptin-activated POMC cells are located more laterally in the ARC than the serotonin-responsive cells [72]. Serotonin modulates the endogenous release of both agonists and antagonists of the melanocortin receptors, which are a core component of the CNS circuitry controlling body weight homeostasis. It should be noted that non-homeostatic or hedonic feeding can override this homeostatic pathway and result in overeating and obesity. Therefore, preventing hedonic feeding by food choices may be beneficial for weight homeostasis.

A sophisticated murine study reported the presence of GABAergic neurons in the dorsal vagal complex as a new player in governing feeding behavior [8]. However, it was reported that GABAergic neurons do not appear to express AgRP and reduce inhibitory tone to postsynaptic POMC neurons [73]. Vong’s study actually challenges the recent sophisticated studies in mice and insinuates that these studies might be flawed. The authors of murine models have shown that GABAergic neurons inhibit NPY [8] or food intake [7]. Moreover, the effects of GABAergic signals are not limited to NPY/AgRP inhibition. GABAergic signals also inhibit (POMC) neurons during fasting [69].

Both anorexigenic neurons as well as orexigenic neurons could be inhibited, or stimulated, depending on the energy status of the host [69]. Therefore, a multitude of feeding possibilities exists even without accounting for the brain region involved. The probability of a non-exclusive event involving multiple molecules with two possible functions (inhibited or stimulated) is impossible to estimate in animal studies. The key point is that GLP-1 and GIP receptor agonists pharmacologically manipulate the neural signals and, consequently, obese individuals lose appetite as well as body weight.

Peripheral injection of fluorescently labeled liraglutide revealed the presence of the drug in the circum-ventricular organs [74]. In this study, murine brain slices showed that GLP-1 directly stimulates POMC/CART neurons and indirectly inhibits neurons expressing neuropeptide Y (NPY) and the agouti-related peptide (AgRP) via GABA-dependent signaling [74]. The labeled liraglutide was internalized in the neurons expressing proopiomelanocortin (POMC) and the cocaine- and amphetamine-regulated transcript (CART) [74].

Beck also reported that the physiological activation of neuropeptides is dependent on the energy availability of the host [69,75]. When energy is deprived or restricted, NPY is activated and, when energy availability returns to normal, NPY synthesis returns to baseline level. Also, NPY metabolism is regulated by diet type, especially carbohydrate and fat content [75]. More recent findings support the view that hypophagia is mediated by GLP-1 receptors in the brain [76].

Toda summarizes these facts: “arcuate melanocortin neurons consist of two distinct neuronal populations: (POMC)-expressing neurons and NPY/AgRP)-expressing neurons” [70]. However, it should be noted that POMC cells are activated by energy surfeit (excess) signals and inhibited by energy deficits [69]. In other words, the energy excesses or deficits are the drivers of POMC cells. So, if the person’s energy state is the driver of POMC cell activation, then the energy state of humans should be controlled to activate or inhibit the POMC cells. We need to pause here to remind ourselves that, in a string of biologic actions, the earliest event is the cause. Therefore, food intake and the resultant energy state of the host are the cause of POMC cell activation. Our view is supported by Beck and Rau, who state that “POMC cell stimulation or inhibition is the consequence of food intake and the resultant energy state” [69,75].

The pharmacological incretin agonism manipulates these neural axes and controls the person’s appetite. However, once these neural inhibitions are lifted, weight regain can occur [25,51]. After the discontinuation of tirzepatide, the study participants regained almost 2/3 of their lost weight within four weeks [51]. We suggest a strategy to mitigate the weight regain after discontinuation of incretin agonists in the FUTURE DIRECTIONS section. These mechanisms and neurohormonal pathways are illustrated in Figure 1.

## 3. Challenges in Incretin Agonism

### 3.1. Involvement of β-Arrestins

As stated in Section 1, incretin and glucagon receptors (GCGR) belong to the class B GPCR, and their efficacy is limited by β-arrestins [77]. β-arrestin-1 (β1arrs) and β-arrestin-2 (Β2arrs) are best known for their ability to mediate the desensitization and internalization of GPCRs [78]. The effect of B2arrs on insulin secretion is dominated by GLP-1R- and no clear GCGR-dependent effects are observed [77].

Β2arrs are ubiquitously expressed and function as negative regulators of GPCR signaling by inhibiting GPCR and G protein coupling, via uncoupling cyclic AMP (cAMP)/protein kinase A (PKA) signaling at physiological levels of GLP-1 [79]. This is a process called desensitization, which dampens insulin secretion. Next, β2-arres scaffold enzymes, phosphodiesterase and diacylglycerol kinase, degrade second messengers generated by G protein activity. This solidifies the desensitization process [80].

At pharmacological doses, the activation of extracellular Thi signal-related kinase (ERK)/cAMP-responsive element-binding protein (CREB) requires Β2arrs. Also, GIP-dependent insulin secretion needs Β2arrs in human islets [79].

β2arrs’ involvement is a potential drawback of incretin agonism because it causes a plateau in the drug efficacies. Avoiding Β2arrs-dependent GPCR desensitization may alleviate the problem of the tachyphylaxis of the GLP-1 and GIP agonism [81]. The recent development of tirzepatide is based on the previous study that revealed that islets β-arrestin1 limits the insulin response of GLP-1, but not GIP, thus may not affect insulin secretion by GIP [82].

The potential inhibition of Β2arrs-dependent GPCR desensitization can be achieved using small interfering RNA [83,84] or blocking beta2 adrenergic receptors which are negative modulators. [85,86]. Unfortunately, GLP-1 and glucagon agonism stimulates the hypothalamic–pituitary–adrenal (HPA) axis which activates beta2 adrenergic receptors and elevates sympathetic tone. Thus, the newer co-agonists of GLP-1 and glucagon may not prevent β-arrestin activation.

### 3.2. β-Cell Exhaustion and Failure

Continuing the stimulation of pancreatic β-cells may cause receptor downregulation and desensitization. A good example of this phenomenon is the insulin resistance in T2D patients [87]. Forcing the β-cells to secrete insulin when they are overwhelmed by insulin demand may accelerate β-cell failure, as shown with sulfonylurea administration [88]. Simply put, chronic agonism eventually reduces the number of receptors, leading to a condition similar to failure of function [89,90]. The mechanism of β-cell failure presumes that the overstimulation of β-cells increases cytoplasmic Ca^++^ levels, and persistent elevation of cytoplasmic Ca^++^ may trigger the apoptosis of β-cells [91]. Under chronic agonism, some β-cells fail to differentiate properly and lose their identity [92].

While these examples describe the overstimulation of β-cells by hyperglycemia in T2D, chronic agonism by incretins may exert similar effects [88,93,94]. Indeed, animal studies have reported increased pyknotic nuclei (a sign of apoptosis) in β-cells and acinar inflammation in rats given a high dose of exenatide [95].

The prime β-cell failure prevention strategy involves not stimulating β-cells [88] or blocking glucokinase [96]. These remedies suggest that the best strategy may be blunting overeating and avoiding insulin requirement by intense lifestyle modification [97] that minimizes β-cell exhaustion.

## 4. Potential Adverse Effects of Incretin Agonism

Although newer incretins appear to be much safer, increased risks of biliary disease [13], pancreatitis [98], bowel obstruction [99], and gastroparesis have been noted [100]. Also, an increased risk of pancreatic or thyroid cancer has been reported [101,102]. It is plausible that insulin secretion, modulated by incretin agonism, may affect pancreatitis and pancreatic cancer because of their shared pathway. Indeed increased risks for multiple pathologies have been reported [103]. Sodhi et al. [103], using a large health claims data set, reported that the use of GLP-1ra compared with bupropion/naltrexone was associated with a 9-fold increased risk of pancreatitis, a 4.2-fold increase in bowel obstructions, and a 3.7-fold increase in gastroparesis [103]. Gastroparesis may be due to neuroendocrine manipulation of the vagal center in the brain by GLP-1ra [104].

The causes of GLP-1ra-associated pancreatitis are not well defined but are hypothesized as incretin agonism may promote pancreatic hyperplasia, leading to increased pancreatic weight and exocrine duct occlusion [105]. The post-marketing FDA adverse event reporting system (FAERS) database clearly demonstrates that GLP-1 agonists are the leading cause (70.2%) of reported pancreatitis [106]. From several studies, we can surmise that GLP-1 agonists reduce the expression of proinflammatory monocyte chemotactic protein-1 and tumor necrosis factor-alpha [107], and make the pancreas vulnerable to infection. Additionally, stimulating insulin secretion increases the expression of insulin-like growth factor (IGF), which may promote pancreatic ductal mucosa proliferation and occlusion. IGF affects both the endocrine and exocrine function of the pancreas [108]. It has been reported that GLP-1 inhibits pancreatic exocrine secretion via the inhibition of vagal outflow [104]. Taken together, the combination of low pancreatic exocrine function, decreased innate immune cytokines, and increased IGF expression may increase the risk of pancreatitis. However, an incidence of pancreatitis resulting from incretin agonism is relatively rare and only large data sets show significant results.

Lu and colleagues [109] explained the mechanism of gastroparesis as “long-term application of GLP-1ras may also elevate the release of endogenous GLP-2. GLP-2 is a cell-specific growth hormone regulating the growth of the small intestine, colonic villi and crypts, increasing villi height, and reducing antral motility” [109]. This study provides the scientific basis for the intestinal blockage and raises concerns for the long-term use of incretin agonism [109].

Recently several cases of pancreatitis and one fatality associated with tirzepatide [110] have been reported. As more people use this class of medications, more serious adverse events may emerge, especially for tirzepatide with added GIP, which masks the nausea and vomiting [38] and allows the patients to tolerate the medication better and to take it for a longer duration. This may increase the number of serious adverse events.

Additionally, a case of leukocytoclastic vasculitis (LCV) induced by once-weekly subcutaneous semaglutide has also been reported [111]. Although 50% of LCV can be idiopathic, immune dysregulation is the presumed cause. Especially, the complete resolution of the lesions shortly after the discontinuation of semaglutide proves the causal role of semaglutide on LCV [111].

Also, GLP-1R and GCGR agonism may increase heart rate (HR) [57,60,112,113] and sympathetic tone [114,115]. Both glucagon [116] and GLP-1 [117] are known to increase sympathetic tone [118,119]. Although some studies have reported no increase in HR, animal studies suggest that GLP-1 acts directly on the sinus node [58,59].

The mechanism through which glucagon receptor (GCGR) activation leads to increased HR is via adenylyl cyclase activation and increases in 3′,5′-c-AMP production in the myocardium [57]. For this reason, glucagon is used as an antidote for a β-blocker overdose [116]. It is conceivable that glucagon is associated with sympathetic tone because hypoglycemia activates the stress response and the hypothalamic–pituitary–adrenal axis [120,121,122,123].

For the mechanism through which GLP-1R contributes to increased HR, it is reasoned that the preproglucagon neurons that mediate GLP-1-associated anorexia are situated in the brainstem to receive signals of stress. This suggests a potential link with the sympathetic nervous system [117,118]. However, another study has reported that the initial increase in sympathetic tone was reversed by GLP-1R activation via acetylcholine and nitric oxide [124]. Making it more confusing, in a human trial, the addition of liraglutide to exercise appeared to blunt the beneficial effects of exercise on left ventricular diastolic function [55]. Many systematic reviews do suggest cardiac protective effects of incretins [125,126]. However, the absolute benefit size was only a 1.5% decrease [127]. Also, a meta-analysis of the CVD benefits of dulaglutide has shown that all individual studies include a relative risk of 1 (no effect). This means that the results are not statistically significant [128].

## 5. Future Directions

How do we harness the observed beneficial effects of incretin agonism on glucose levels and weight loss, minimizing potential harm? These are the wishes of experts and patients alike. They have all raised the concern about the potential adverse events from the long-term use of incretin agonism, the sustainability of their efficacy, and the high cost [129,130,131]. We concur with the experts and suggest that the remarkable weight loss and glycemic control of incretin agonism should be juxtaposed to the high cost, its unsustainability of efficacy, and the potential adverse effects. As we stated in the neuro-hormonal control of appetite section, manipulating the neuronal axis fails when the manipulation terminates. Thus, tapering GLP-1 agonism and enhancing lifestyle modification may be the key to healthy weight maintenance. It is pivotal to teach the patients so that they can make healthier food choices that maintain satiety [132], and to encourage physical activities. A group in Denmark has done this by slowly tapering semaglutide and applying intense lifestyle modification in a trial [133]. More longer-term studies are needed, emphasizing lifestyle modifications in addition to tapering GLP-1ras.

## Figures and Tables

**Figure 1 cells-13-01842-f001:**
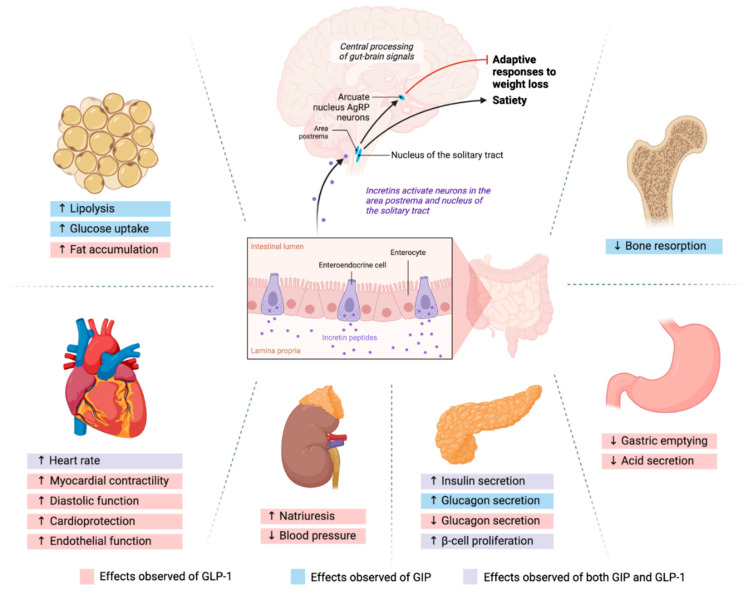
Neuroendocrine axis and incretin actions on multiple organs.

**Table 1 cells-13-01842-t001:** Key randomized trial results on incretin agonism and health outcomes.

First Author, Trial Name/ID, Year, Phase	Sample Size, Duration	Target Population, Methods	Objectives, Results (s), and Comments
Le Roux, [NCT01272219], 2017, Phase 3 [15]	N = 2254, 68 weeks or 160 weeks	Prediabetic obese and overweight with co-morbidities cohort Drug: Lira 3.0 mg or placebo SC injection once daily	Objectives: Weight reduction/maintenance, T2D onset delay. Results: By 160 weeks, DM Dx given to (2%) of 1472 in Lira vs. (6%) of 738 in the placebo. Time to DM diagnosis was 99 wks in Lira vs. 87 wks in placebo.
Husain M, PIONEER 6 [NCT02692716], 2019, phase 3 [16]	N = 3183, Median 62 weeks	T2D with high cardiovascular risk cohort Drug: Sema or placebo Oral administration once daily	Objectives: Cardiovascular safety of oral Sema, QD in T2D patients. Primary outcome: incidence of MACE. Results: MACE occurred in 3.8% in Sema vs. 4.8% in placebo including 15 CVD mort. in Sema vs. 30 in placebo.
Frías JP, AWARD-11 [NCT03495102], 2021, phase 3 [14]	N = 1842, 52 weeks (36 weeks primary endpoint)	T2D Patients inadequately controlled with metformin Drug: Dula 1.5 mg, 3.0 mg, or 4.5 mg SC injection once weekly	Objectives: Change in HbA_1c_ by week 36 from baseline. Results: At 36 wks, Dula 4.5 mg superior to 1.5 mg with [ETD] −0.24% but Tx estimand of 3.0 mg was not significant (*p* = 0.096). However, vomiting nearly doubled in 4.5 mg level. (5.6% vs. 9.3%)
Rubino D, STEP-4 [NCT03548987], 2021, phase 3a [12]	N = 803, 68 weeks	Obese or overweight cohort without T2D Drug: Sema 2.4 mg or placebo SC injection once weekly Primary outcome: Change in body weight (%)	Objectives: Comparison of SC Sema continued or switch to placebo, both with lifestyle intervention. Wt. change week 20–68: Sema −7.9% vs. placebo +6.9%. G-I adverse events: Sema 49.1% vs. placebo 26.1% (1.88 times more in Sema group).
Frías JP, SURPASS-2 [NCT03987919], 2021, phase 3 [18]	N = 1879, 40 weeks	Metformin-treated T2D cohort Drug: Sema 1 mg or Tirzep 5 mg, 10 mg, 15 mg SC injection once weekly	Objectives: Compare effect of Sema and Tirzep on blood sugar levels. Outcome = Change in HbA_1c_ by week 40. The diff. between groups Tirzep 5-mg, 10-mg, and 15-mg, and Sema were −0.15%, −0.39%, and −0.45%, respectively. Serious adverse events: 5–7% in Tirzep vs. 3% in Sema.
Ludvik B, SURPASS-3 [NCT03882970], 2021, phase 3 [23]	N = 1444, 52 weeks	Metformin-treated or metformin with SGLT2i-treated T2D cohort Drug: Tirzep 5 mg, 10 mg, 15 mg, or insulin degludec 100 U/mL (titrated) SC injection once weekly (Tirzep), SC injection once daily (insulin degludec)	Objectives: Assess safety and efficacy of Tirzep versus insulin degludec on blood sugar levels. Results: Non-inferiority of Tirzep to insulin. HbA1c change in Tirzep 5, 10, 15 mg at week 52 were −1·93%, −2·20%, and −2·37%, respectively, and −1.34% in insulin. G-I adverse events: 7% in Tirzep vs. 1% in insulin group. Hypoglycemia: 4% in Tirzep vs. 7% in insulin gr.
Del Prato S, SURPASS-4 [NCT03730662], 2021, phase 3 [24]	N = 2002, 52 weeks (treatment continued until maximum 104 weeks)	Metformin-treated, sulfonylurea-treated, SGLT2i-treated T2D cohort Drug: Tirzep 5 mg, 10 mg, 15 mg, or glargine 100 U/mL (titrated) SC once weekly (Tirzep), SC once daily (glargine)	Objective: Assess efficacy and safety of Tirzep versus insulin glargine in adults with high CVD risk and T2D. Primary outcome: Non-inferiority of Tirzep 10 mg or/and 15 mg versus glargine. Mean HbA_1c_ change at week 52: −2·43% and −2·58%, with Tirzep 10, 15 mg, respectively, vs. −1·44% with glargine.
Rubino DM, STEP-8 [NCT04074161], 2022, phase 3b [13]	N = 338, 68 weeks	Obese or overweight cohort without T2D Drug: Sema 2.4 mg or Lira 3.0 mg or placebo (matching for both conditions) SC injection once weekly (Sema), SC injection once daily (Lira)	Objectives: Assess the efficacy of once-weekly Sema vs. once-daily Lira on weight loss. Change in body weight (%) by week 68. Mean wt. change from baseline: −15.8% with Sema, −6.4% with Lira, and −1.9% with placebo. G-I adverse events: 84.1% with Sema, 82.7% with Lira.
Wilding J, STEP 1-extension [NCT03548935], 2022 [25]	N = 327, 1 year after withdrawal from STEP-1	Extension analysis Previous drug: Sema 2.4 mg or placebo	Objectives: body weight changes and cardio-metabolic factors following Sema withdrawal. Primary outcome: One year after withdrawal of weekly Sema 2.4 mg + lifestyle intervention, participants regained two-thirds of their prior weight loss.
Heerspink H, SURPASS-4 Post Hoc Analysis, 2022 [17]	N = 2002, Median 85 weeks (104 weeks max)	Metformin-treated, sulfonylurea-treated, SGLT2i-treated T2D cohort Drug: Tirzep 5 mg, 10 mg, 15 mg, or glargine 100 U/mL (titrated) SC injection once weekly (Tirzep), SC injection once daily (glargine)	Objectives: Compare the effects of Tirzep and insulin glargine on the kidney. Primary outcome: Tirzep slowed the eGFR decline (1.4 vs. 3.6 mL/min) and UACR increased with insulin while with Tirzep it decreased by −6.8% compared with insulin glargine.
Dahl D, SURPASS-5 [NCT04039503], 2022, phase 3 [26]	N = 475, 40 weeks	T2D with titrated insulin glargine on glycemic control cohort Drug: Tirzep 5 mg, 10 mg, 15 mg, or placebo SC injection once weekly	Objectives: Assess efficacy and safety of Tirzep in T2D patients receiving inadequate glycemic control. Primary outcome: Mean changes in HbA_1c_ were −2.40%, −2.34%, and −0.86% with 10 mg, 15-mg Tirzep, and placebo, respectively.
Lincoff AM, SELECT [NCT03574597], 2023, phase 3 [27]	N = 17,604, Mean 137 weeks (Mean follow up 160 weeks)	Obese or overweight cohort with CVD and without T2D Drug: Sema 2.4 mg or placebo SC injection once weekly	Objectives: Assess reduction in risk of having cardiovascular events. Primary outcome = MACE (CVD mortality + nonfatal MI + nonfatal stroke). MACE 6.5% in Sema, 8.0% in placebo (Risk Diff. = 1.5%). SAE leading to permanent discontinuation was doubled in Sema. (16.6% in Sema, 8.2% in placebo).
Jastreboff AM, [NCT04881760], 2023, phase 2 [22]	N = 338, 48 weeks	Obese or overweight with weight-related comorbidities cohort without T2D Drug: Reta 1 mg, 4 mg, 8 mg, 12 mg, or placebo SC injection once weekly Retatrutide = multireceptor agonist of (GLP-1 + GIP + glucagon)	Objectives: Assess efficacy of Reta on body weight loss. Primary outcome: Change in body weight (%) by week 24. Results: Wt. change at 24 weeks −7.2% (1-mg), −12.9% (4-mg), −17.3% (8-mg), and 17.5% in the 12-mg retatrutide groups, −1.6% placebo. HR peaked at 24 weeks and declined thereafter. NB: Comparator should have been Tirzep, not placebo, to show adding glucagon would be safe.
Aronne L, SURMOUNT-4 [NCT04660643], 2024, phase 3 [28]	N = 670, 88 weeks (36 weeks onward placebo could be administered)	Cohort: Obese or overweight without T2D Drug: Tirzep 10 mg, 15 mg Mean wt. loss 20.9%. At week 36, randomized to continue Sema or placebo.	Objectives: Assess Tirzep effect on maintenance of body weight reduction. Primary outcome: Mean change in weight from week 36 until week 88 (%). Results: Those switched to placebo group regained 14% wt. (67%), those continuing tirzepatide lost an additional 5.5%.
Loomba R, [NCT04166773], 2024, phase 2 [29]	N = 190, 52 weeks	Cohort: Confirmed-MASH with liver fibrosis Drug: Tirzep 5 mg, 10 mg, 15 mg, or placebo SC injection once weekly	Objectives: Assess safety and efficacy of Tirzep as a MASH treatment. Primary outcome: Resolution of MASH without worsening of fibrosis by week 52. Results: Risk diff. 34%, 46%, and 53% at Tirzep 5-mg, 10-mg, and 15-mg, respectively.
Sanyal AJ, [NCT04771273], 2024, phase 2 [30]	N = 293, 48 weeks (24 weeks rapid-dose phase, 24 weeks maintenance phase)	Confirmed-MASH with fibrosis cohort Drug: Survo 2.4 mg, 4.8 mg, 6.0 mg, or placebo SC injection once weekly	Objectives: Assess safety, tolerability, and efficacy of Survo (dual agonist of glucagon and GLP-1 ra) as a MASH treatment. Primary outcome: Reduction in MASH with no worsening of fibrosis by week 48. Results: Risk diff. of liver fat decrease were 49%, 53%, and 43% in the 2.4 mg, 4.8 mg, and 6.0 mg Survo groups, respectively. More nausea in Survo (66% vs. 23%), diarrhea (49% vs. 23%), and vomiting (41% vs. 4%); SAEs were similar.
Sanyal AJ, [NCT04881760], 2024, phase 2a [31]	N = 98, 48 weeks	Obese or overweight with weight-related comorbidities cohort without T2D Drug: Reta 1 mg, 4 mg, 8 mg, 12 mg, or placebo SC injection once weekly	Objectives: Assess safety, tolerability, and efficacy of Reta for body weight loss, assess liver fat at 24 weeks. Results: At 24 weeks, normal LF was achieved by 27%, 52%, 79%, and 86% with 1 mg, 4 mg, 8 mg, and 12 mg of Reta and 0% (placebo). LF reductions were related to changes in wt., abdominal fat, and metabolic measures of insulin sensitivity.

Nota Bene: In real-world data, age at the time of GLP-1 initiation and GLP-1 ra cessation increased the risk of major adverse cardiac events (MACE) [32].These facts strongly suggest that age and other factors could be important confounders.

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
