# Peer review of "Does Incretin Agonism Have Sustainable Efficacy?"

_cells, 2024, doi:10.3390/cells13221842_

Round 1
Reviewer 1 Report
Comments and Suggestions for Authors
The title of this manuscript makes one believe that they will discuss longevity of effect of the GLP-1 related receptor agonists. That is misleading. There is no discussion of the long term studies. The authors delve into the area of neural effects, and that is worthwhile since, more and more, the satiety benefits of these agents appear to be the predominant effect. If the authors want to expand on this subject, it would be useful, but they would have a significant amount of research to pursue to bring this to fruition.
Reviewer 2 Report
Comments and Suggestions for Authors
Comments and suggestions about the author will be displayed in the PDF document

The language of the essay is fluent, grammatically accurate, and the vocabulary is used appropriately. Can be modified to fit the content of your essay
Reviewer 3 Report
Comments and Suggestions for Authors
The study titled “Incretin agonism: Sustainable efficacy or surreptitious hazard?” offers a thorough discussion of the mechanisms by which GLP-1, GIP, and amylin regulate insulin and glucagon secretion, effectively highlighting their distinct roles in metabolic regulation. However, I recommend several areas for further discussion to enhance the manuscript.
1. Delayed gastric emptying is a common side effect of GLP-1 receptor agonists in both humans and mice. However, recent studies have reported no significant effect on gastric emptying in humans, which contradicts earlier findings. The authors should provide a more detailed discussion of this discrepancy, addressing potential reasons for the divergent outcomes and their implications for clinical understanding.
2. Further discussion on potential interventions to prevent β-cell failure during long-term incretin therapy is needed.
Round 2
Reviewer 1 Report
Comments and Suggestions for Authors
The authors raise a valid issue as to sustainability of GLP-1 receptor agonist effect. There have been no true log term studies. Inasmuch as the GLP-1 receptor agonists have become the primary treatment for obesity and are now used to treat morbid obesity, the question of long term sustained benefit is a major issue. If the authors could focus on that question and delve into what is know and what is not known abut duration of effect, they could make a useful contribution.
Author Response
Reviewer 1, Round 2 Comments:
The authors raise a valid issue as to sustainability of GLP-1 receptor agonist effect. There have been no true log term studies. Inasmuch as the GLP-1 receptor agonists have become the primary treatment for obesity and are now used to treat morbid obesity, the question of long term sustained benefit is a major issue. If the authors could focus on that question and delve into what is know and what is not known abut duration of effect, they could make a useful contribution.
Our response: We agree with reviewer 1 that assessing the long-term benefit as well as harm is valid issue. We created a subchapter dedicated to long-term effects in the manuscript and clearly stated what has been known and misrepresented at the present time. Here is what we have added:
“””
1-d. Long-term efficacy of incretin agonism
As we stated in the abstract, the long-term effects of incretin agonism are unknown at present. This is because there are no long-term clinical trials and the longest-term randomized trial is of 160-week (40 month). As the GLP-1ras have become the leading treatment for obesity and diabetes, the long-term efficacy is a critical issue given the high cost and potential harm. We, as well as other scientists, raised the same issue, that there is no evidence for long-term sustainable efficacy of GLP-1ras.[61]
Although several extension studies are published, these are not randomized trials. Rather, they are bias-prone observational studies. Nevertheless, these extension studies gave us a glimpse of long-term effects of incretin agonism. The longest extension study is DURATION-1 where exenatide once weekly (QW) was given subcutaneously and followed for 7 years.[62] The drop-out rate of this extension study was 59% and this is almost 3 times higher than the acceptable 20% drop-out rate.[63] The weight loss was attenuated to just 3.9% at the 7th year. This is quite important because Moll et al. reported that at least 10% body weight loss is necessary to observe salient benefit over harm, while 5% weight loss did not show any net benefit.[64] The harm in these analyses are mostly gastrointestinal (G-I) adverse effects. In a meta-analysis, Zhang et al. observed that the adverse G-I effects were due to neurohormonal suppression of appetite center in the brain.[65]
Additionally, the clear trend of attenuation in efficacy was observed with increasing study duration. Mean HbA1c reduction was 1.7%, 1.6% and 1.5% at 2nd, 5th, and 7th year, respectively. In the 3rd year, only 28%, and in the 7th year, 46% of the study subjects achieved HbA1c 7%. Seemingly improving HbA1c levels in the 7th year was due to high drop-out rate: in 3rd year, the cohort had 194 subjects, which decreased to 122 in the 7th year. The DURATION-1 authors [62] stated that some CVD markers improved in DURATION -1. However, this is irrelevant to GLP-1 agonism because the study participants were taking several CVD medications in addition to GLP-1 agonists. Thus, improved CVD markers cannot be attributed to GLP-1 agonism. One positive comment was that the adverse effects were similar to the randomized portion of the study.[62]
“””